# Sedentary Behaviour—A Target for the Prevention and Management of Cardiovascular Disease

**DOI:** 10.3390/ijerph20010532

**Published:** 2022-12-28

**Authors:** Abbie C. Bell, Joanna Richards, Julia K. Zakrzewski-Fruer, Lindsey R. Smith, Daniel P. Bailey

**Affiliations:** 1Institute for Sport and Physical Activity Research, School of Sport Science and Physical Activity, University of Bedfordshire, Bedford MK41 9EA, UK; 2Centre for Physical Activity in Health and Disease, Brunel University London, Uxbridge UB8 3PH, UK; 3Division of Sport, Health and Exercise Sciences, Department of Life Sciences, Brunel University London, Uxbridge UB8 3PH, UK

**Keywords:** sedentary behaviour, cardiovascular disease, prolonged sitting, cardiovascular risk markers

## Abstract

Cardiovascular disease (CVD) is highly prevalent and can lead to disability and premature mortality. Sedentary behaviour, defined as a low energy expenditure while sitting or lying down, has been identified as an independent risk factor for CVD. This article discusses (1) the association of total sedentary time and patterns of accumulating sedentary time with CVD risk markers, CVD incidence and mortality; (2) acute experimental evidence regarding the acute effects of reducing and breaking up sedentary time on CVD risk markers; and (3) the effectiveness of longer-term sedentary behaviour interventions on CVD risk. Findings suggest that under rigorously controlled laboratory and free-living conditions, breaking up sedentary time improves cardiovascular risk markers in individuals who are healthy, overweight or obese, or have impaired cardiovascular health. Breaking up sedentary time with walking may have the most widespread benefits, whereas standing breaks may be less effective, especially in healthy individuals. There is also growing evidence that sedentary behaviour interventions may benefit cardiovascular risk in the longer term (i.e., weeks to months). Reducing and breaking up sedentary time may, therefore, be considered a target for preventing and managing CVD. Further research is needed to determine the effectiveness of sedentary behaviour interventions over the long-term to appropriately inform guidelines for the management of CVD.

## 1. Introduction

Cardiovascular disease (CVD) is an umbrella term used to describe diseases associated with the heart and blood vessels, such as coronary heart disease (CHD), peripheral arterial disease and stroke [1]. It is estimated that CVD causes 17.9 million deaths per year, making it the most prevalent non-communicable disease globally [2]. It was estimated that 168,000 years of life were lost in 2020 in the UK as a result of CVD, in addition to causing 4598 disability adjusted life years [3]. A major cause of CVD is atherosclerosis, which is characterised by fatty-fibrous lesions that develop slowly over many years [4]. These lesions occur as a result of chronic oxidative stress and increased inflammation that cause damage to the artery walls (endothelium) [5]. Over time, these lesions can progress into an atheroma with a thin fibrous cap. Disruption of this plaque can result in an aneurysm leading to platelet aggregation and thrombus formation [4]. If the thrombus occludes the coronary artery for a prolonged period, it causes a reduction in myocardial perfusion and cardiovascular complications, such as ischaemia [4]. It is important to understand the factors that contribute to the aetiology of CVD so that effective interventions can be identified to address this major cause of mortality and disability.

The progression of atherosclerosis and incidence of CVD is largely attributable to haemodynamic, chemical and behavioural risk factors such as high blood pressure, increased low-density lipoprotein cholesterol (LDL-C), high glucose, increased adiposity, smoking and low levels of physical activity (defined as any bodily movement resulting in energy expenditure) [6,7,8,9]. Sedentary behaviour refers to a low energy expenditure (≤1.5 METs) while in a reclined, lying or seated posture during waking hours [10]. Technological advances and changes in society have led to significant increases in sedentary time in the general population. A rapidly growing volume of research in recent years has identified sedentary behaviour as a distinct risk factor associated with CVD incidence and mortality [8,11,12,13,14,15,16]. This review, therefore, considers sedentary behaviour as a target for the prevention and management of CVD in the context of:a)The association of total sedentary time and patterns of accumulating sedentary time with CVD risk markers, CVD incidence and mortality;b)Experimental evidence regarding the acute effects of reducing and breaking up sedentary time on CVD risk markers; and,c)The effectiveness of interventions targeting sedentary behaviour on CVD risk.

## 2. Associations of Sedentary Behaviour with Cardiovascular Disease Risk Markers

### 2.1. Total Sedentary Time

There is a growing evidence base of observational studies exploring the associations of sedentary behaviour with CVD risk markers [7,17]. Findings from the Nord-Trondelag Health (HUNT) study demonstrated that self-reported sitting ≥10 h per day was significantly associated with adverse CVD risk marker levels (body mass index (BMI), waist circumference, systolic blood pressure (SBP), diastolic blood pressure (DBP), triglycerides and non-fasting glucose) compared with sitting <4 h per day [18]. In support of these findings, the Australian Diabetes, Obesity and Lifestyle (AuSDiab) study found that in a sample of 2103 men and 2751 women, there was a significant deleterious association of higher total sitting time with waist circumference, BMI, SBP, triglycerides, high-density lipoprotein cholesterol (HDL-C), 2 h plasma glucose and fasting insulin [19]. The associations reported in these studies were independent of moderate-to-vigorous physical activity (MVPA), suggesting that sedentary time may be a distinct risk factor associated with CVD risk. However, a key limitation of this evidence is the measurement of sedentary time using self-report methods, which can lead to significant underestimations of sedentary behaviour [20]. The determination of sedentary behaviour using the IPAQ questionnaire is indicative, as participants declare the time they believe that they spend sitting [21]. A comparison of the International Physical Activity Questionnaire (IPAQ) measure of total sitting and the activPAL activity monitor (considered the gold-standard method for measuring sitting time [22]) found that the IPAQ underestimated total sitting time by 3.4 h per day [20]. This may lead to underestimations of the true magnitude of association between sedentary time and CVD risk markers. Hence, estimating sedentary time using device-based methods is important to improve understanding in relation to the associations of this behaviour with CVD risk [23].

Studies in which sedentary time has been measured using device-based methods have observed significant adverse associations of increasing sedentary time with CVD risk markers. In a healthy sub-sample of the AusDiab study (*n* = 169), there were significant adverse associations of total sedentary time, measured using a hip worn accelerometer, with waist circumference and a clustered metabolic risk score, after adjusting for confounders including MVPA [24]. Similarly, a cross-sectional analysis of baseline data collected from two intervention studies in the UK, comprising of a young adult population (*n* = 153, aged 32.9 ± 5.6 years) and a middle to older age population (*n* = 725, aged 63.7 ± 7.8 years) with increased risk of developing Type 2 diabetes mellitus (T2DM), found that accelerometer-assessed sedentary time was significantly adversely associated with glucose tolerance, triglycerides and HDL-C [25]. Similar to self-report evidence, the associations in these studies were independent of MVPA. Consistent findings have also been found in larger cohort studies, such as the National Health and Nutrition Examination Survey (NHANES 2003-06) which included a large ethnically diverse sample of 4757 participants [26]. Increasing total daily sedentary time, measured using an ActiGraph accelerometer, was significantly adversely associated with waist circumference, fasting triglycerides and fasting insulin [26]. It could be argued that the most convincing evidence supporting the theory that sedentary time is adversely associated with cardiovascular risk is from meta-analyses that pool data from the available evidence to provide greater statistical power. A meta-analysis of 46 studies including 70,576 participants aged 18–87 years reported that increasing device-measured total sedentary time was associated with higher fasting glucose, fasting insulin, triglycerides, waist circumference and lower HDL-C [7].

The evidence in this field is not consistent, though, in relation to the adverse associations of sedentary time with CVD risk markers being independent of physical activity. Maher et al. [27] found that the adverse association of total sedentary time with CVD risk markers became non-significant after adjusting for total physical activity. This study was unique to many other investigations as adjustment was for both light physical activity and MVPA. Previous accelerometer-based studies have typically only adjusted for MVPA [25,26]. Sedentary time and physical activity may, therefore, not be independent from one another with respect to cardiovascular risk markers. Instead, associations that are attenuated by adjusting for physical activity may represent a displacement mechanism, whereby the adverse associations of increasing total sedentary time with CVD risk markers are a result of reduced physical activity, and vice versa. Isotemporal substitution modelling of NHANES data has shown that reallocating 30 min per day of sedentary time with an equal volume of light intensity physical activity or MVPA was associated with lower insulin, insulin resistance and lipid levels [28]. It may thus be appropriate to target replacing sedentary time with physical activity, as opposed to concentrating exclusively on reducing total sedentary time, per se.

Based on the studies reviewed here, there appears to be relatively consistent evidence to suggest that higher total sedentary time is adversely associated with CVD risk markers. The findings also appear to be consistent with respect to this association being independent of physical activity levels, especially MVPA. This suggests that high levels of physical activity may not eliminate the increased CVD risk marker levels that are associated with higher volumes of sedentary time. The adverse associations of total sedentary time with CVD risk markers appear to be stronger and more consistent in studies using device-based measures. It is thus recommended that future research employs device measures of sedentary time where possible to provide a precise understanding of the associated cardiovascular health risks.

### 2.2. Sedentary Time Bouts and Breaks

In addition to total sedentary time being adversely associated with CVD risk markers, there is evidence to suggest that sedentary bout durations and the number of breaks in sedentary time (i.e., a period of sedentary time interrupted by a bout of standing or physical activity) may be important to consider for cardiovascular health. In the NHANES survey, a higher number of breaks in sedentary time per day (measured using accelerometery) was significantly beneficially associated with fasting insulin, waist circumference and C-reactive protein [26]. These associations were independent of total sedentary time and MVPA, suggesting that individuals may need to limit both the total daily amount of sedentary time in addition to regularly breaking up bouts of sedentary time [26]. Findings from the Northern Finland Birth Cohort 1966 study (*n* = 4439) found that, after adjustment for total sedentary time and MVPA, participants who engaged in shorter sedentary bouts (durations of <15–30 min; termed “Shortened sitters”) had significantly improved 2 h insulin, fasting serum insulin, triglycerides, total cholesterol/HDL-C ratio and LDL-C/HDL-C ratio compared to those who engaged in less daily sedentary bouts and had a shorter average sedentary bout duration [29]. These findings were supported by data that more frequent interruptions in sedentary time (measured using accelerometery) were beneficially associated with triglycerides and glucose tolerance, independent of MVPA, total sedentary time and the average intensity of the physical activity breaks [30]. Likewise, in individuals recruited as part of the 2016–2018 wave of the 1970 British Cohort Study, for every additional 10 breaks in sitting time measured using the activPAL device, there was a significant reduction in BMI, body fat percentage and HbA1c [31]. These data support a beneficial association of regularly breaking up sedentary time with CVD risk markers.

The evidence regarding the associations of breaks in sedentary time with CVD risk markers is not consistent, though. A longitudinal study over 6 months found that the number of breaks in sedentary time at baseline was associated with waist circumference but not HDL-C fasting insulin or insulin resistance in individuals with newly diagnosed T2DM [32]. Inconsistent evidence was also reported in a meta-analysis demonstrating that breaks in sedentary time were not associated with glucose metabolism or cardiovascular health. There was, however, beneficial associations of increased breaks with BMI and waist circumference [33]. Potential reasons for the discrepancies across studies may be due to inconsistencies in how the number, duration and intensity of breaks is determined and considered in the analysis. Furthermore, the majority of these studies measured sedentary time using devices that were unable to identify posture, meaning that standing may be misclassified as sedentary time. This may lead to inaccurate estimates of the number of breaks in sedentary time and the associations with CVD risk markers.

There is evidence to suggest that the number of breaks in sedentary time could be targeted to improve cardiovascular health. However, the causal effects of manipulating breaks in sedentary time cannot be inferred from these studies. Thus, the effects of breaking up sedentary time on CVD risk markers in experimental studies is an important focus of research in this field and is considered later in this review.

## 3. Sedentary Behaviour and Cardiovascular Disease Incidence and Mortality

Arguably, the deleterious associations of increased sedentary behaviour with CVD incidence and CVD-mortality were first identified in the landmark bus worker study by Morris [34]. A significantly lower CVD rate was seen in male ticket conductors than sedentary bus drivers, with an overall annual incidence of 1.9/1000 people in ticket conductors, versus 2.7/1000 people in sedentary bus drivers. It was also found that postal workers who walked or cycled to deliver post had significantly less cardiovascular events each year (1.8 per 1000 people per year) compared to their more sedentary colleagues (telephonists and civil service clerks) who experienced 2.4 events per 1000 people per year. Postal workers also had a 50% lower rate of CVD mortality [34]. Sedentary behaviour, as a distinct health risk behaviour, received little attention in the following decades. It was not until the early part of this century that researchers began to further explore the potential relationship of sedentary behaviour with CVD and mortality.

### 3.1. TV Viewing

The exponential growth in research exploring the associations of sedentary behaviour with CVD incidence and mortality may have stemmed from studies exploring associations of TV viewing with health outcomes. One of the earlier studies to investigate this was the AusDiab study [35], with baseline measures taken between 1999 and 2000. Television viewing was self-reported over seven days by 8800 participants, with a median follow up of 6.6 years. For each 1 h increment of daily TV viewing, there was an 11% increased risk of all-cause mortality (hazard ratio (HR) 1.11, 95% CI 1.03, 1.20) and 18% increased risk of CVD mortality (HR 1.18, 95% CI 1.03, 1.35), after adjusting for covariates including exercise time [35]. The European Prospective Investigation into Cancer (EPIC) Norfolk prospective study followed 12,608 healthy participants for 6.9 ± 1.9 years during which 2620 individuals developed CVD (2134 of these events were non-fatal) [35]. Participants who developed CVD watched TV for approximately 0.5 h/day more at baseline than those who did not develop CVD. After adjustment for covariates, including total physical activity energy expenditure, a HR of 1.06 (95% CI 1.03, 1.09; *p* < 0.001) was found for non-fatal cardiovascular events with each additional hour/day of TV viewing [36]. In support of this, Stamatakis [37] reported that, after adjusting for MVPA, >2 h of self-reported TV viewing time per day was associated with a HR of 1.98 (95% CI 1.09, 3.59) for CVD events in 4512 individuals over a mean follow up period of 4.3 years. Nonetheless, a meta-analysis which pooled data from eight studies found an increased risk for each additional 2 h of TV viewing time per day for fatal and non-fatal CVD (HR 1.15; 95% CI 1.06, 1.23) and all-cause mortality (HR 1.13; 95% CI 1.07, 1,18). While the association between TV viewing and CVD incidence was linear, the risk of all-cause mortality appeared to increase when the duration of TV viewing time was greater than 3 h per day [13]. The research showing that TV viewing was adversely associated with CVD incidence and mortality led to an abundance of research in the following years investigating other types of sedentary behaviour and total daily sedentary time in this context.

### 3.2. Total Sedentary Time and Independence from Physical Activity

Total daily sedentary time has been identified as a potential intervention target for the prevention and management of CVD in a growing number of studies published in this area. As such, a number of meta-analyses have been conducted to better understand the evidence regarding the risk of CVD incidence and mortality associated with increased sedentary time. Based on data from 18 studies in 794,557 participants, Wilmot [8] found a pooled HR of 1.71 (95% CI 1.08, 2.48) for cardiovascular mortality and 2.47 (95% CI 1.44, 4.24) for CVD incidence in participants who self-reported the highest duration of sedentary time compared to those who engaged in the least amount, after adjusting for physical activity. Another meta-analysis reported a significant pooled HR of 1.29 (95% CI 1.27, 1.30) for CVD incidence in 448,285 participants pooled from nine studies who engaged in the highest amount of daily sitting compared to the least. This association was attenuated, but remained significant, after adjustment for physical activity (HR 1.14; 95% CI 1.04, 1.23) [11]. This suggests that the association of sedentary time with CVD may be partially mediated by physical activity. A meta-analysis of 34 prospective design studies including 1,331,468 participants observed a non-linear relationship for total sedentary time and CVD-mortality that was independent of physical activity levels [14]. There was a non-significant relative risk (RR) for CVD-mortality of 1.01 (95% CI 0.99, 1.02) for each additional hour of sedentary time in participants who engaged in ≤6 h of sedentary time per day, but a significant RR of 1.04 (95% CI 1.03, 1.04, *p* < 0.001) for each additional hour in individuals who were sedentary for >6 h per day [14]. These findings suggest that the risk of CVD may be greater in individuals who engage in higher amounts of daily sedentary time regardless of their physical activity level.

Despite growing evidence from meta-analyses that the association of sedentary behaviour with CVD incidence and mortality may be independent of physical activity [8,11,14], these studies have not typically stratified participants according to their sedentary time and physical activity levels. Thus, such research cannot adequately address whether higher physical activity levels may attenuate or eliminate the detrimental health risks associated with higher sedentary time. To address this, a meta-analysis with a median follow up of 10.2 years stratified 850,060 participants into quartiles of physical activity and daily sitting to examine the joint associations with CVD mortality [15]. There was a dose–response association for higher sitting and CVD mortality in the least active participants. Yet, for the second and third quartiles of physical activity, the association of increased sitting with CVD mortality was less consistent and this association was entirely eliminated in the most active participants [15]. These findings suggest that targeting reductions in sitting may be most important in individuals who engage in low amounts of physical activity, whereas high physical activity levels may offer protection against the increased CVD mortality risk associated with higher sitting.

## 4. Effects of Breaking up Sedentary Time on Cardiovascular Risk Markers

Informed by observational findings that total sedentary time and breaks in sedentary time are associated with cardiovascular health, research in this field progressed to investigating the causal effects of reducing and breaking up sedentary time on CVD risk markers in controlled laboratory-based studies. These studies have primarily focused on the acute effects of breaking up prolonged sitting on postprandial glucose, insulin and lipids, which are strong predictors of future CVD risk [38,39]. This is because excess postprandial elevations in glucose, insulin and lipids cause increased oxidative stress and inflammation resulting in endothelial dysfunction and atherosclerosis over time [40].

### 4.1. Effects of Breaking up Sedentary Time on Postprandial Glucose and Insulin

There is consistent evidence to suggest that breaking up prolonged sitting with short bouts of light or moderate intensity physical activity reduces postprandial glucose and insulin over a single day in a range of population groups, including individuals who are healthy, overweight and obese, have impaired metabolic health, or have T2DM [41,42,43,44,45,46]. A number of studies have found that interrupting sitting with light intensity walking for 2 min every 20 min significantly reduces postprandial glucose by 9–16% when compared to prolonged sitting in healthy individuals [42,47,48]. Reductions were large in a study where light-intensity walking breaks were of a longer duration (20 min every hour), with postprandial glucose being significantly lower by 38% compared to prolonged sitting [41]. Light intensity physical activity breaks have been shown to be equally as beneficial as moderate intensity breaks. For example, Dunstan [43] found similar postprandial glucose responses to interrupting prolonged sitting with either 2 min of light or moderate intensity walking every 20 min in individuals with overweight or obesity [43]. This was supported by findings from a meta-analysis of 20 cross-over design studies in apparently healthy individuals, which demonstrated that regular physical activity breaks significantly reduced postprandial glucose irrespective of the intensity of the physical activity breaks [44]. The reduction in glucose in response to physical activity breaks that are aerobic in nature may be due to blood glucose being used as fuel for energy during short duration muscular contractions [49]. Based on the findings discussed here, it appears that carbohydrate metabolism related to physical activity breaks could be similar for light and moderate intensity walking.

There are a number of studies in which breaking up sitting has not affected postprandial glucose, but has attenuated postprandial insulin responses. This was observed in a study in Qatari females with mixed weight status where sitting was interrupted with 3 min of moderate intensity walking every 30 min [45]. Similarly, a significant reduction in postprandial insulin was reported when sitting was interrupted by 2 min of moderate intensity walking every 30 min in healthy individuals [50]. Reductions in insulin with no concomitant change in glucose concentration may be explained by an enhancement in the insulin-stimulated glucose disposal pathway induced by an increase in sitting interruptions. A proposed mechanism for this is an increase in insulin sensitivity via upregulation of insulin signalling pathways [50]. Contraction-mediated glucose uptake into skeletal muscle by the glucose transporter protein, GLUT-4, may also be increased [50]. It could therefore be hypothesised that less insulin is synthesised via the insulin-mediated pathway, as there is a secondary pathway acting upon postprandial glucose. However, breaking up sitting with longer duration, less frequent bouts (10 min of moderate intensity walking every 180 min) did not affect postprandial glucose or insulin [51]. Sitting may, therefore, need to be interrupted more frequently to impart beneficial effects on postprandial glucose and insulin.

The effects of simple resistance exercise breaks as a potentially feasible alternative to walking or standing breaks have also been investigated. One study found that 3 min bouts of resistance exercise every 30 min led to similar reductions in postprandial glucose and insulin compared with light intensity walking in individuals with T2DM [52]. However, interrupting sitting with 5 min of simple resistance exercises every 30 min in young healthy individuals led to significantly higher postprandial glucose compared with prolonged sitting [53]. A potential mechanism explaining this is a shift from carbohydrate to fat metabolism during resistance exercise in healthy individuals [54]. Therefore, resistance breaks appear to be more beneficial in those with impaired cardiovascular health than healthy individuals, although this requires further investigation.

With standing breaks potentially being more practical and attractive than physical activity (e.g., through allowing office workers to continue working whilst standing), there has been interest in the acute effects of breaking up sitting with standing. Yet, findings are inconsistent. In healthy participants, breaking up sitting with 2 min of light intensity walking every 20 min significantly reduced postprandial glucose over 5 h, but breaking up sitting with 2 min of standing every 20 min did not [48]. Similar findings have been reported in other studies with apparently healthy participants when sitting was interrupted with 2 to 5 min of standing every 20 to 30 min [42,55]. Contrary to this, in healthy, inactive males, intermittent standing significantly reduced postprandial glucose over 9 h, with beneficial effects persisting into the following day [56]. Standing bouts were, however, longer in duration (15 min every 30 min) compared to studies reporting no effect of standing. Another reason for the difference between studies may be because participants were inactive in the study by Benatti [56], meaning that the standing bouts may have provided a larger stimulus to evoke metabolic improvements relative to more active populations [48]. Therefore, standing bouts of longer duration and higher volume may be needed to impart glucose benefits in healthy individuals. Nonetheless, 45 min standing bouts every hour also did not lead to acute improvements in postprandial glucose in healthy individuals [57]. The lack of response may have been due to postprandial responses only being measured the day after the standing breaks had been performed, by which time any effects may have dissipated [58]. The effects of standing on postprandial insulin in healthy individuals appear to have been seldom reported, with no apparent significant effects seen [56]. This should be addressed in future research to appropriately inform the potential effects of breaking up sitting with standing on postprandial metabolism in this population group.

Contrary to the findings that breaking up sitting with standing has limited effects on postprandial glucose and insulin in healthy individuals, there is more consistent evidence that standing breaks are beneficial in individuals who are overweight, obese or have impaired cardiometabolic health. For instance, compared to prolonged sitting, postprandial glucose was significantly lower in response to standing (−39%) walking (−24%) and cycling breaks (−44%) that progressed from 10 to 30 min in duration over 8 h in participants with overweight and obesity [59]. In a simulated work day, office workers with overweight or obesity who interchanged between sitting and standing every 30 min also exhibited a significant reduction in postprandial glucose compared to a day of seated work [60]. Standing breaks of shorter durations (5 min every 30 min) were also sufficient for reducing postprandial glucose and insulin in post-menopausal women with dysglycaemia [61]. To summarise the available experimental data, a meta-analysis of seven acute, 1-day randomised crossover trials supported the findings of these individual studies. It was found that breaking up sitting with standing had a moderate effect on reducing postprandial glucose in participants who were predominantly overweight or obese [58].

Experimental evidence suggests that breaking up prolonged sitting with light or moderate intensity physical activity significantly improves postprandial glucose and insulin in a range of population groups. Individuals with impaired cardiometabolic health appear to benefit more from interrupting prolonged sitting with standing breaks compared to healthy individuals. Breaking up sitting may, therefore, be an appropriate intervention target for reducing CVD risk, especially in individuals with overweight, obesity or impaired cardiometabolic health.

### 4.2. Effects of Breaking up Sitting on Postprandial Lipids

A number of studies evaluating the effects of accumulating physical activity in short regular bouts across the day on postprandial lipids have found that interrupting sitting with moderate-intensity physical activity reduced postprandial triglyceride responses the following day [62,63]. Such reductions in lipids the next day have been seen in healthy individuals who accumulated 3 min light intensity walking breaks every 30 min or 2 min moderate intensity walking breaks every 30 min when compared with prolonged sitting [63]. Similarly, triglycerides were attenuated the next day when individuals with obesity engaged in 3 min bouts of moderate intensity cycling every 30 min [64]. These findings were supported by subgroup analyses within a meta-analysis, where pooled data demonstrated that regular breaks in sitting attenuated triglycerides, but only 12–16 h later, with no changes occurring on the day that the physical activity breaks were performed [44]. A hypothesised mechanism explaining the attenuated triglyceride response to physical activity breaks is an increase in cellular regulation of lipoprotein lipase (LPL) activity as a result of avoiding prolonged periods of muscular inactivity [65]. The activity of LPL may peak at around 8 to 22 h post-exercise [66], which would explain why breaking up sitting has benefits in postprandial triglycerides the next day.

In contrast to meta-analysis data that the benefits of breaking up sitting appeared to occur primarily the following day [44], there is growing evidence that breaking up sitting with physical activity of at least moderate-intensity may impart benefits on the day that the physical activity breaks take place. Breaking up sitting with 3 min moderate intensity walking breaks every 30 min significantly reduced postprandial triglycerides in Qatari females with a high mean body fat level [45]. However, when prolonged sitting was interrupted with 8 min moderate intensity cycling breaks every hour, there was no difference in triglycerides compared to prolonged sitting [67]. This non-response could be explained by the lack of weight bearing and localised muscular contractions or infrequent nature of the physical activity breaks, meaning an insufficient stimulus for LPL activity. Indeed, other studies with more frequent whole-body physical activity (i.e., walking) have seein reductions in postprandial triglycerides. This includes interrupting sitting with 90 s moderate intensity walking breaks every 15 min [68] and 1.5 min of moderate intensity walking approximately every 25 min [69]. Interrupting sitting with high intensity physical activity has also led to attenuations in postprandial lipids. High intensity walking/jogging breaks for 2 min 32 s every hour across 8 h led to significant reductions in postprandial triglycerides and increases in HDL-C in healthy individuals on the day that the breaks took place [70]. However, another study in which high intensity interval cycling (60 s at 90% maximum oxygen uptake) was spread across the day occurring every 30 min, postprandial lipids were unaffected [71]. The lack of response may be due to the physical activity (i.e., cycling) being localised to the lower body and not being weight-bearing, which may limit the potential stimulus of muscular activity on LPL activity. Breaking up sitting with moderate or high intensity physical activity may thus have potential to improve postprandial lipids over the course of the day that the breaks take place if they are frequent enough and involve whole-body muscular contractions.

In contrast to evidence showing benefits of moderate and high intensity physical activity breaks on postprandial lipids, a number of studies have found that light-intensity walking breaks (3-to-5 min breaks every 30 min) are ineffective [48,55,61]. Similar to the findings of postprandial glucose, standing may also appears to provide an insufficient stimulus to evoke reductions in postprandial lipids. In healthy individuals, interrupting prolonged sitting with standing for 15–45 min each hour over a single day did not affect postprandial triglycerides the day after in healthy individuals [57]. There was also no effect in healthy individuals from breaking up sitting with standing for 2 min every 20 min when triglycerides were measured on the day that the breaks occurred [48]. In individuals with impaired metabolic health (dysglycemic, post-menopausal women) and older adults, postprandial triglycerides were also not significantly affected during the day when sitting was interrupted with 5 min of standing every 30 min [55,61]. A greater physical activity stimulus involving dynamic muscular contractions of at least moderate intensity may be needed for breaks in sitting to impart beneficial effects on postprandial lipids.

### 4.3. Effects of Breaking up Sitting on Blood Pressure

Breaking up sitting appears to have potential for reducing resting blood pressure over a single day in individuals with varying health statuses. One study found that breaking up sitting with 20 min of light intensity walking each hour significantly reduced SBP and DBP in healthy participants [41]. Similarly, in young, healthy males, a significant reduction in SBP was found when prolonged sitting was interrupted with 3 min of light intensity walking every 30 min over 6.5 h, compared to uninterrupted sitting [63]. In individuals who were overweight or obese, Wheeler [72] found that breaking up sitting with light intensity walking significantly reduced SBP. Despite benefits being seen across groups with respect to their health status, the benefits may be greatest in those with impaired metabolic health. Breaking up sitting with light intensity walking or simple resistance exercise breaks for 3 min every 30 min reduced SBP by 14 ± 1 mmHg and 16 ± 1 mmHg, respectively, and DBP by 8 ± 1 mmHg and 10 ± 1 mmHg, respectively [73]. This is substantially greater than the reductions reported in other studies which ranged from 3–8 mmHg [41,72]. The greater reductions in the study by Dempsey [73] may have been due to 17% of the participants being pre-hypertensive and 71% being hypertensive, suggesting greater potential for responses to breaks in sitting compared to normotensive individuals. These findings show that regardless of health status, light intensity physical activity breaks have a significant acute beneficial effect on resting blood pressure, but the benefits may be greatest in individuals with higher blood pressure levels.

The benefits of breaking up sitting on blood pressure may be seen with a variety of physical activity modalities. Interrupting sitting with progressively longer intervals of standing, light intensity walking or cycling (10–30 min breaks every hour over an 8.5 h day) in individuals with overweight or obesity each significantly reduced SBP compared to prolonged sitting [74]. However, the evidence in this field is not consistent as a number of studies have reported null effects when breaking up sitting with light or moderate intensity physical activity [48,67,75,76]. Two studies in healthy participants adopted a longer duration (8–10 min) but less frequent (every 60 min) standing [77] or moderate intensity cycling breaks [67]. This could suggest that more frequent sitting interruptions are needed to improve blood pressure in healthy individuals. However, both Bailey and Locke [48] and Bhammar [75] interrupted sitting with 2 min of light or moderate intensity walking breaks every 20 min, but this did not affect blood pressure. The study by Bailey and Locke [48] was limited as blood pressure was only measured at two time points, whereas the study by Bhammar [75] measured ambulatory blood pressure that included measurements taken in close proximity to the walking bouts, which could have influenced the outcome. Further research is needed to establish the potential influence of different frequencies and durations of breaks in sitting on blood pressure and should also seek to standardise when measurements are taken to eradicate this as a potential reason for discrepancy in findings. There is also a lack of literature regarding the effects of breaking up sitting with high intensity physical activity on blood pressure, which should be addressed in future studies.

There is data to suggest that the type of physical activity used to break up sitting may influence blood pressure responses. A systematic review found that ‘aerobic interruption strategies’ consisting of walking, running or cycling led to reductions in SBP and DBP, whereas simple resistance exercises and standing were not effective [77]. This may suggest that the mechanisms through which breaking up sitting reduces blood pressure are stimulated only with higher intensity (relative to standing) or whole-body physical activity. A potential mechanism by which breaking up sitting with ‘aerobic’ based physical activity affects blood pressure includes attenuating the reduction in vascular sheer stress that is caused by prolonged sitting [78]. Previous laboratory-based studies have identified that prolonged sitting leads to a reduction in blood flow and sheer stress due to a reduced muscular and metabolic demand [79,80]. A reduction in shear stress leads to vasoconstriction of the blood vessels and endothelial dysfunction, that can result in higher blood pressure [81]. Physical activity breaks may also reduce sympathetic nervous system activity that could be a result of the break in sitting or secondary to reductions in circulating insulin [73]. Further studies are needed to confirm these hypothesised mechanisms, including consideration of population-level studies to better understand the associations of breaks in sedentary time with vascular health, such as shear stress.

Overall, breaking up sitting appears to significantly reduce blood pressure in healthy individuals and those with overweight, obesity or impaired cardiovascular health. However, there is limited data to suggest sitting breaks that incorporate standing or simple resistance activities are effective. Further research is needed to affirm the suitability of standing and resistance exercise breaks in reducing blood pressure.

## 5. Effects of Reducing and Breaking up Sitting in Free-Living Conditions

As discussed above, there is an abundance of evidence to suggest that breaking up sitting can improve cardiovascular risk markers under rigorously controlled laboratory conditions [48,56], but these studies lack external validity. To address this issue, several studies have explored the effects of reducing and breaking up sitting under free-living conditions. In healthy, sedentary individuals, substituting 6 h of sitting with a combination of 2 h standing and 4 h of light intensity walking (activity regimen) over four days under free-living conditions significantly attenuated postprandial insulin during an oral glucose tolerance test, compared with a day of predominantly sitting (<1 h of walking and 1 h of standing each day; sitting regimen) [82]. Fasting triglycerides and non-HDL cholesterol (measured the morning after each 4 day activity regimen) were also significantly reduced by the activity regimen compared to the sitting regimen [82]. In participants with overweight or obesity, four days of sitting, standing and light walking for 7.6 h, 4.0 h and 4.3 h per day (SitLess), respectively, were compared to four days of sitting for 13.5 h, 1.4 h of standing and 0.7 h of light-intensity walking per day (Sit) [83]. Significant reductions in fasting insulin, triglycerides and non-HDL cholesterol, and significant increases in HDL-C were found in the SitLess arm [83]. However, these studies were limited to metabolic biomarkers being measured only on the day after the 4-day regimens. The temporal changes in CVD risk markers throughout the regimen periods were thus unknown. Continuous glucose monitoring (CGM) has, therefore, been utilised in subsequent free-living study designs to assess temporal glucose dynamics throughout each day of the monitoring period.

In individuals with T2DM, replacing 4.7 h of sitting per day with 2.2 h of walking and 2.5 h of light intensity stepping for four consecutive days significantly reduced 24 h continuous glucose concentrations compared to prolonged sitting [84]. Yet, no differences in free-living continuous glucose responses over four days of reduced total daily (−58 min/day) and prolonged sitting (−99 min/day), and increased stepping time (+40 min/day), were reported in individuals with overweight and obesity [85]. Similarly, a study in which individuals with obesity were reminded by a smartwatch to interrupt their sitting with 3 min of light physical activity every 30 min over 3 weeks did not improve average 24 h glucose curves or glucose tolerance [86]. Fasting glucose and intra-day glucose variability did, however, significantly improve [86]. These contradictory findings suggest that improvements in continuous glucose may only be realised in individuals with a deteriorated metabolic health status (i.e., individuals with T2DM). Alternatively, greater volumes of sitting may need to be replaced with standing or walking for benefits in continuous glucose to occur. That said, 24 h glucose was unaffected in participants with T2DM who interrupted their sitting with 5 min of walking for 2 h after each postprandial period [87]. Null responses in this study could be due to participants being instructed to maintain their habitual activity in the comparison (control) arm compared to higher imposed volumes of sitting in the study by Duvivier [84]. This led to a smaller difference in sitting, standing and stepping between the conditions in the study by Blankenship [87]. The findings of these studies suggest that, like rigorously controlled laboratory settings, reducing and breaking up sitting in free-living conditions has potential for improving cardiovascular health in the short-term, particularly in those with impaired cardiovascular health. It is important that longer-term studies are conducted to evaluate the effectiveness of interventions targeting reductions and breaks in sitting to further support the use of these strategies in optimising cardiovascular health.

## 6. Interventions to Reduce Sedentary Behaviour and Improve Cardiovascular Health

A number of studies have evaluated the effectiveness of interventions targeting reductions in total and prolonged sitting on CVD risk markers. The majority of this literature has focused on office workers as this occupational setting is inherent of high sitting [88]. A short-term fully powered randomised controlled trial (RCT) evaluating a multi-component intervention that included education, a step challenge, telephone support and prompts to break up sitting in sedentary office workers found no significant changes in daily workplace sitting over eight weeks [89]. However, time spent in prolonged bouts of sitting was reduced significantly by 39 min per shift in the intervention group compared with controls. This was alongside significant increases in breaks from sitting by 7.8 per shift and stepping time by 12 min per shift [89]. Although waist circumference was significantly reduced by 1.6 cm in the intervention group relative to controls, there were no improvements in other CVD risk markers including BMI, body fat %, blood pressure and lipids [89]. The reasons for the limited effects on CVD risk markers in this study may include the short duration of the intervention, which may not be sufficient for imparting chronic adaptations in physiologic and metabolic processes that would benefit cardiovascular health. Conversely, an e-health intervention that passively prompted office workers to stand or engage in light activity during working hours over 13 weeks significantly improved mean arterial pressure compared to controls [90]. Interventions using sit-stand desks, desk pedal devices and education have also reported improvements in some CVD risk markers and no change in others [91]. A systematic review examining the effects of sedentary behaviour interventions (ranging from 4 weeks to 13 months in duration) on cardiovascular risk markers in office workers found that 20 out of 29 included studies observed a significant improvement in at least one cardiovascular risk marker (such as SBP, fasting glucose and HDL-C) in response to the intervention [91]. However, many of these studies were limited by non-randomised designs, short durations and not being sufficiently powered to detect changes in cardiovascular risk markers [91].

The ‘Stand Up Victoria’ study addressed the above limitations of previous research in a fully powered RCT that evaluated a multi-component workplace intervention including organisational (e.g., senior management support), environmental (e.g., height-adjustable desk) and individual components (e.g., goal setting) in 231 office workers over 12 months. Workplace sitting and prolonged sitting were significantly reduced by 45.4 min and 40.1 min per 8 h workday, respectively, after 12 months [92]. Sitting time was predominantly replaced by standing, likely due to the ability to complete standing work using a height-adjustable workstation. At 12 months, fasting glucose and a composite cardiovascular risk score were maintained in the intervention group, but significantly deteriorated in the control group [93]. The Physical Activity at Work (PAW) study found that a multi-component workplace intervention consisting of individual (pedometers and financial incentives), social (group movement breaks), environmental (posters with suggestions to reduce sitting) and organisational (encouragement from leaders) components did not reduce occupational or daily sedentary time in 282 office workers over 6 months [94]. This may have been due to the intervention not including provision of height-adjustable workstations, meaning that participants had limited opportunity to stand whilst working. Cardiovascular risk marker changes have not yet been reported for the PAW study so it is unclear whether the non-response in sedentary time may have led to deteriorations in cardiovascular health. The evidence suggests that interventions to reduce and break up sitting may improve cardiovascular risk markers in office workers and, therefore, should be considered for reducing population risk of CVD.

There is limited evidence regarding the effectiveness of sedentary behaviour interventions for managing cardiovascular health in individuals with CVD and T2DM. Of the limited studies to date evaluating sedentary behaviour interventions in individuals with cardiometabolic disease, a randomised controlled feasibility study in individuals with T2DM evaluated the use of a smartphone app that had features to support reducing and breaking up sitting (e.g., goal setting and alerts to prompt movement) [95]. After eight weeks, participants reported the app to be acceptable and the intervention appeared to have potential efficacy for reducing total daily and prolonged sitting and increasing the number of breaks in sitting per day. Body fat percentage and glucose tolerance also appeared to improve [95]. The definitive effects of this intervention require evaluation in a definitive RCT to inform its appropriateness for managing cardiovascular health in this population group. There are studies ongoing evaluating the effects of reducing and breaking up sitting in individuals with T2DM and CVD. This includes the OPTIMISE your health multi-component intervention for office workers with T2DM [96]. The intervention consists of health coaching, a height-adjustable workstation and an activity tracker taking place over 18 months, with primary outcomes including overall daily sitting and HbA1c.

Similarly, the ROSEBUD (Reduced Occupational Sedentary BehavioUr in type 2 Diabetes) study is assessing the effects of an intervention comprising of a physical activity tracker wristband, text message reminders and nurse-led counselling over 12 months on cardiometabolic outcomes in office workers with T2DM [97]. There is also an ongoing feasibility trial evaluating the REgulate your SItting Time (RESIT) intervention for reducing and breaking up sitting in individuals with T2DM [98]. The RESIT intervention is delivered remotely (not within a workplace setting) and includes an online education programme, health coaching, and self-selected smartphone apps, computer-prompt software, and wearable devices. In individuals with coronary artery disease who are undergoing a cardiac rehabilitation programme, the SIT LESS study is evaluating the effects of a personalised secondary prevention intervention (patient education, goal setting, motivational interviewing with coping planning and telemonitoring using a pocket worn physical activity tracker) on sedentary behaviour and quality of life [99]. These studies will provide important evidence regarding the feasibility, acceptability and effectiveness of sedentary behaviour interventions that may inform clinical care guidelines for these population groups with a focus on reducing and breaking up sedentary time.

## 7. Conclusions

This review has provided a narrative overview of evidence in relation to reducing and breaking up sedentary behaviour as a target for the prevention and management of CVD. Observational evidence appears to be consistent with regard to increased sedentary time being adversely associated with CVD incidence and mortality; these associations may also be independent of physical activity. There is also literature that demonstrates increased daily sedentary time is detrimentally associated with cardiovascular risk factors, whereas an increased number of breaks in sedentary time is beneficially associated with these outcomes. A large body of evidence suggests that breaking up sitting within controlled laboratory settings can improve postprandial glucose, insulin, triglycerides and blood pressure, especially in participants with impaired metabolic health. It also seems to be the case that individuals with impaired metabolic health may benefit from light intensity physical activity breaks, whereas apparently healthy individuals may benefit more consistently only when the physical activity breaks are of a moderate or high intensity. There are some studies to suggest that reducing and breaking up sitting under free-living conditions has potential for improving cardiovascular health in the short-term. Longer-term intervention studies that reduce and break up sitting may improve cardiovascular risk markers, but the quality of evidence in this area is weak. Reducing and breaking up sedentary behaviour can be considered for the prevention and management of CVD, although sufficiently powered longer-term RCTs are required to provide definitive outcomes regarding the chronic effects of sedentary behaviour interventions, especially in population groups with cardiometabolic disease.

## Data Availability

Not applicable.

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
