# Peer review of "Sedentary Behaviour—A Target for the Prevention and Management of Cardiovascular Disease"

_ijerph, 2022, doi:10.3390/ijerph20010532_

Round 1
Reviewer 1 Report
Dear Authors
I commend your narrative review that explored the impact of sedentary behavior and its interruptions on the cardiometabolic markers (BMI, WC, serum biomarkers). The flow of sections that narrates about sedentary behavior on risk markers, interruptions on risk markers and dose-response effects on markers are really intriguing. The authors have appropriately addressed the HUNT study, AusDiab, and protocols of ROSEBUD, RESIT studies.
However, few concerns need to be addressed:
1. I feel the sections could be rearranged: (1) sedentary behavior and its impact on markers; (2) realloaction with physical activity on markers; (3) sedentary behavior interruptions on markers; (4) dose response relation of physical activity or sedentary behavior interruptions. The contents are good
2. 95th line and 147th line: 2- hr glucose? or glucose tolerance - kindly clarify
3. 118 line - "This should be explored further in future studies" - Already studies have explored the total sedentary time with markers independent of physical activity, Newer studies have established favorable cardiometabolic responses when reallocating SB with PA. 1. Sjöros, T., Vähä-Ypyä, H., Laine, S. et al. Both sedentary time and physical activity are associated with cardiometabolic health in overweight adults in a 1 month accelerometer measurement. Sci Rep 10, 20578 (2020). https://doi.org/10.1038/s41598-020-77637-3
4. 176th line - 1.9/1000vs 2.7/1000 respectively?
5. Section 3.3 - daily sedentary time??? topic is not linking the section. I would suggest to change. 'inspite of physical activity'
6. 256th line - 34th and 35th references are narrative reviews. Please add emprical evidence on acute effects of interruptions on metabolic risk
7. 370th line - introducing dose response effects of sedentary behavior . I feel this aspect is vital important and needs a seperate section?
8. 456th line - please consider revising "Shear stress? I guess the postulated mechanism is "no reduction in shear stress as observed with acute experimental trials that administered prolonged sitting in lab based trials. But population based trials remain inconsistent"
9. 546th line - The author quoted favorable effects of Standup Victoria trial only. The null effects of sitting breaks on metabolic risk in "PAW study" needs to be addressed
10. 28th and 29th references are repeated
11. 54th reference needs to be formated as per the journal guidelines
12. 47th and 67th references are repeated - please consider revising

Reviewer 2 Report
Dear authors you take up a very important research topic, the work is coherent and has a good scientific output but it needs supplementation
introduction:
line 54- the authors write a huge amount of research and only 2 items of literature are given (8) (11) Please expand this section and add literature from 2022-2022.
2.1
please add and note that the determination of sedentary behavior using the IPAQ questionnaire is only indicative. Respondents subjectively declare the time they spend sitting. Please expand on this thesis and support it with literature
4.1
The authors describe the role of resistance exercise as having a beneficial effect on metabolism used as breaks from sitting. Please supplement this section with other types and forms of exercise and their effects on carbohydrate metabolism or reducing insulin resistance.
4.2
line 376-377 since as the authors state based on the literature benefits with 8 minutes of moderate intensity cycling exercise, please add the results of studies supported by the literature where high intensity efforts were used. Are there any studies supporting the effectiveness of such interventions based on high intensity or submaximal efforts. Please also add here information on physical activity used as a high intensity sedentary interval such as high interval training (HIT) efforts
4.3
In this section, please also add studies showing that an active break can include high-intensity efforts and what benefit this can have on blood pressure values in both healthy and sick individuals.
Conclusions
Please also note in the vignettes the possibility of using interventions based not only on low and moderate intensity efforts
